# Association of Risk-Taking Behaviors, Vestibular Provocation and Action Boundary Perception Following Sport-Related Concussion in Adolescents

**DOI:** 10.3390/brainsci15030229

**Published:** 2025-02-22

**Authors:** Shawn R. Eagle, Anthony P. Kontos, Shawn D. Flanagan, Christopher Connaboy

**Affiliations:** 1Department of Neurological Surgery, University of Pittsburgh, Pittsburgh, PA 15213, USA; 2Department of Orthopaedic Surgery, University of Pittsburgh, Pittsburgh, PA 15213, USA; 3Center for Lower Extremity Ambulatory Research (CLEAR), Rosalind Franklin University, Chicago, IL 60064, USA; shawn.flanagan@rosalindfranklin.edu (S.D.F.); christopher.connaboy@rosalindfranklin.edu (C.C.)

**Keywords:** perception–action coupling, Perception–Action Coupling Task (PACT), concussion, adolescents, vestibular symptoms/impairments, risk taking

## Abstract

**Background/Objectives**: The purpose of this study was to evaluate the association between risk-taking behaviors, vestibular symptoms/impairment and perception–action coupling behavior in recently concussed adolescents. **Methods**: This study utilized a cross-sectional design to evaluate the early effects of concussion on 12–18-year-old adolescents (n = 47) recruited from a concussion specialty clinic at their presenting clinical appointment. The Perception–Action Coupling Task (PACT) was used to assess action boundary perception by evaluating the participant’s ability to quickly and accurately determine whether a virtual “ball” fits in a virtual “hole”. Accuracy, response time and inverse efficiency were evaluated at the 0.8 and 1.2 ratios of ball–hole pairings, where 0.8 indicates the ball was slightly smaller than the hole and 1.2 indicates the ball was slightly larger than the hole. The Balloon Analog Risk Task (BART) is a computerized test which measures risk-taking behavior by “pumping” up a balloon. Each pump provides a small amount of virtual money into their bank; the goal is to make as much money as possible without popping the virtual balloon. The Vestibular Ocular Motor Screening (VOMS) tool is a brief screening tool designed to identify ocular or vestibular dysfunction following sport-related concussion, where horizontal/vertical vestibular ocular reflex (VOR) and visual motion sensitivity (VMS) are the primary vestibular outcomes. Pearson correlation matrices were developed to evaluate the association between BART, VOMS and PACT outcomes within the study cohort of concussed adolescents. **Results**: PACT inverse efficiency at the 1.2 ball–hole ratio was significantly correlated with all three VOMS outcomes (r = 0.33–0.37). The standard deviation of pump reaction time during BART was significantly correlated with accuracy (r = −0.47) and inverse efficiency (r = 0.42) at the 1.2 ratio. The standard deviation of the total number of pumps during BART was significantly correlated with PACT response time at the 1.2 ratio (r = 0.34). Horizontal VOR correlated with balloons collected (r = −0.30) and balloons popped (r = −0.30). **Conclusions**: The results of this study suggest that risk-taking behaviors and vestibular symptoms/impairment are associated with worse action boundary perception in adolescents following concussion. This relationship is more pronounced in male adolescents than females.

## 1. Introduction

Sport-related concussion remains a significant public health concern for United States adolescents, as up to 2 million occur each year [1]. Recent work has demonstrated an increased risk for another concussion and/or musculoskeletal injury within the first year of returning from the index concussion [2]. No clear causal mechanism has emerged linking concussion and subsequent musculoskeletal injury within the first few months of returning to activity, but poor sensorimotor control and/or premature return to activity from concussion have been postulated [3,4]. In a scoping review, Eagle et al. [4] suggested that a combination of the patient’s unique post-concussion response (i.e., vestibular symptoms, migraine, mood disruption) and risk factors for these primary symptom drivers (e.g., history of motion sickness, biological sex, mood disorders) could alter perception-action coupling and increase risk for subsequent injury. Perception–action coupling is based upon direct perception theory which argues that the person interacting with their environment can directly perceive opportunities for action from environmental cues and an inherent attunement to their own physical capabilities [5]. Trauma to the head could alter a person’s perceptions of their environment and/or their attunement to their own abilities, both of which could factor into increasing subsequent injury risk [4].

Intrinsic behaviors could also contribute to altered perception–action coupling, as dysregulated impulse control or higher risk-taking patterns contribute to increased injury risk [6]. Prior work has shown that after sport-related concussion, adolescents had higher cognitive instability and attention-based impulsivity on the Barratt Impulsiveness Scale compared to non-concussed adolescents [6]. Moreover, cognitive instability and attention impulsivity were associated with perception–action coupling performance in multivariable modeling amongst the concussed adolescents [6]. The term “labyrinthine concussion” has been used to describe disruption to the vestibular system after head trauma [7]. Vestibular symptoms and impairment, such as dizziness, imbalance, vertigo and gait dysfunction, which are among the most commonly affected domains following concussion, have also not been examined in relation to perception–action coupling. Researchers have shown that disrupted vestibular function can increase risky decision making in non-concussed individuals, which may, in turn, influence perception–action coupling performance [8]. As such, it is important to note that pre-existing vestibular dysfunction could be related to increased injury risk from a greater likelihood of falls. However, the link between vestibular symptoms/impairment following concussion and perception–action coupling has not yet been examined empirically in at risk-adolescents. The purpose of this study was to evaluate the association between risk taking behaviors, vestibular symptoms/impairment and perception–action coupling behavior in recently concussed adolescents. We hypothesize that more risk taking will be associated with higher vestibular symptoms and worse perception–action coupling performance. We also hypothesize that higher vestibular symptoms will be associated with worse perception–action coupling performance.

## 2. Materials and Methods

### 2.1. Study Design and Participants

This study utilized a cross-sectional design to evaluate the early effects of concussion on 12–18-year-old adolescents (n = 47) recruited from a concussion specialty clinic at their presenting clinical appointment. Sport-related concussion was diagnosed per international consensus criteria, which include reporting a mechanical force transmitted to the brain from either direct or indirect impacts resulting in signs and/or symptoms of brain injury [9]. Participants were included if presenting to the concussion specialty clinic for evaluation within 21 days of injury and had not yet received clearance for return to play. Participants were excluded if they reported a previous diagnosis of neurological/vestibular disorder, a history of brain injury or moderate–severe traumatic brain injury and/or a current psychiatric disorder. The University of Pittsburgh Institutional Review Board approved the study for human subject research (ID number: PRO18060287; approved 6 September 2018). Participants provided written informed consent prior to participating in study procedures.

### 2.2. Measures

Perception–Action Coupling Task (PACT)

The PACT was used to assess action boundary perception by evaluating the participant’s ability to quickly and accurately determine whether a virtual “ball” fits in a virtual “hole” [10]. The participant reacts to a virtual presentation of a ball–hole pairing by moving their finger from the home button to the joystick. Participants are instructed to move the joystick forward if the ball fits in the hole. Participants are instructed to move the joystick backward if the ball does not fit in the hole. For the present analysis, the accuracy of decisions, response time and inverse efficiency were evaluated. Response time was recorded from the moment the participant’s finger was removed from the home button until the joystick has been moved. Inverse efficiency is the combination of speed and accuracy into a single measure where higher scores indicate a less efficient performance (i.e., slower responses and more errors) and lower scores indicate more efficient performance (i.e., faster responses and accurate responses). Accuracy, response time and inverse efficiency were evaluated at the 0.8 and 1.2 ratios of ball–hole pairings, where 0.8 indicates the ball was slightly smaller than the hole and 1.2 indicates the ball was slightly larger than the hole. These ratios were the closest in this version of the PACT to the actual action boundary (i.e., a ratio of 1.01 where the ball would only be slightly larger than hole). The PACT has good intersession reliability (intraclass correlation coefficients = 0.78–0.94), within-subject variability [10] and population validity in patients with concussion [6,11].

Vestibular Ocular Motor Screening (VOMS) Tool

The VOMS is a brief screening tool designed to identify ocular or vestibular dysfunction following sport-related concussion. Participants self-report headache, dizziness, nausea and fogginess symptom severity prior to testing from 0 to 10. Of the seven VOMS subtests, three are primarily evaluations of the vestibular system: horizontal/vertical vestibular ocular reflex (VOR) and visual motion sensitivity (VMS). Following each subtest, participants self-report headache, dizziness, nausea and fogginess symptom severity from 0 to 10 again to evaluate any symptom provocation the test may have had from baseline. VOR is conducted by having the participant hold a tongue depressor straight out from their face and rotating their head 50 degrees side-to-side to the beat of a metronome while maintaining visual contact with the tongue depressor for a total of 20 repetitions. VMS is conducted by having the participant hold a tongue depressor straight out from their face and rotating their upper body and arm 80 degrees side-to-side while maintaining visual contact with the tongue depressor. Horizontal and vertical VOR and VMS were recorded as change scores from post test to baseline. If there was no change in symptoms after testing, the score was recorded as 0. The VOMS has a good test–retest reliability (0.60–0.81) and internal consistency (Cronbach alpha = 0.97) with population validity in the concussed patient [12,13,14].

Balloon Analog Risk Task (BART)

The BART is a computerized test which measures a person’s risk-taking behavior (Psychology Software Tools, Inc.; Sharpsburg, PA, USA) [15]. Participants are asked to “pump” up a balloon and each pump provides a small amount of virtual money into their bank. The goal is to make as much money as possible without popping the virtual balloon. The balloons randomly pop at different volumes, and the probability of it exploding increases with each pump. If the balloon explodes, the money accumulated from pumping is lost. BART outcomes are the number of balloons collected, the number of balloons popped, the total money collected, the mean and standard deviation of the total pumps, and the mean and standard deviation of the reaction time to press the pump button. More balloons collected/fewer pumps reflects a more conservative strategy, whereas more balloons popped/more pumps reflects a more aggressive strategy. The BART has good test–retest reliability (intraclass correlation coefficient = 0.77) [16] and population validity in participants with concussion [17].

### 2.3. Statistical Analysis

This is a secondary analysis for which the original study was not statistically powered a priori. Data were assessed for normality using the Shapiro–Wilk test. Descriptive statistics were calculated for all outcome measures, including means and standard deviations for continuous variables. All continuous variables were compared with independent samples *t*-tests between male and female participants and categorical variables were compared with Fisher’s Exact Test. Pearson correlation matrices were developed to evaluate the association between BART, VOMS and PACT outcomes within the study cohort of concussed adolescents. Because male adolescents take risks more frequently than female adolescents [18], correlation matrices were also stratified by biological sex assigned at birth. Alpha level was set a priori to *p* < 0.05 but those with *p* < 0.01 are also reported. Correlation strength was interpreted as 0.10–0.39 = weak correlation, 0.40–0.69 as moderate correlation, 0.70–0.89 as strong correlation and 0.90–1.00 as very strong correlation [19]. Statistical analyses were conducted with the Statistical Package for the Social Sciences (IBM, Armonk, NY, USA; v29).

## 3. Results

Descriptive statistics can be viewed in Table 1. On average, participants were 15.2 ± 1.8 years old, presented to the concussion specialty clinic 9.5 ± 6.0 days from injury and reported a history of 1.5 ± 0.8 concussions. Over 70% of the participants were male (n = 33). When comparing male to female participants on all outcome measures, there were no statistically significant differences between groups except that males reported a slightly higher total number of prior concussions (*p* = 0.03).

Correlations between PACT outcomes with VOMS and BART outcomes for the total concussed cohort can be viewed in Table 2. PACT inverse efficiency at the 1.2 ball–hole ratio had a significant positive correlation with all three VOMS outcomes (r = 0.33–0.37; *p* = 0.01–0.03). The standard deviation of pump reaction time during BART was negatively correlated with accuracy (r = −0.47; *p* < 0.001) and positively correlated with inverse efficiency (r = 0.42; *p* = 0.004) at the 1.2 ratio. The standard deviation of the total number of pumps during BART was positively correlated with PACT response time at the 1.2 ratio (r = 0.34; *p* = 0.03). No other statistically significant associations were observed.

Correlations between VOMS and BART outcomes for the total concussed cohort can be viewed in Table 3. Horizontal VOR negatively correlated with balloons collected (r = −0.30; *p* = 0.047) and balloons popped (r = −0.30; *p* = 0.047). No other statistically significant associations were observed.

Correlations between PACT, VOMS and BART outcomes for the male and female concussed adolescents can be viewed in Table 4. For male participants, PACT accuracy at the 1.2 ratio negatively correlated with horizontal VOR (r = −0.42; *p* = 0.02), vertical vestibular ocular reflex (r = −0.42; *p* = 0.02) and VMS (r = −0.44; *p* = 0.01). PACT inverse efficiency at the 1.2 ratio positively correlated with VOMS outcomes, as well (r = 0.46–0.52; *p* = 0.003–0.009). PACT accuracy at the 1.2 ratio negatively correlated with balloons collected (r = −0.36; *p* = 0.04) and standard deviation of pump reaction time (r = −0.47; *p* = 0.005) and negatively correlated with balloons popped (r = 0.36; *p* = 0.04). Inverse efficiency at the 1.2 ratio also positively correlated with the standard deviation of pump reaction time (r = 0.47; *p* = 0.006). No statistically significant associations were observed for female participants.

Correlations between VOMS and BART outcomes for male and female concussed adolescents can be viewed in Table 5. For male participants, horizontal VOR and VMS positively correlated with balloons collected (r = 0.37; *p* = 0.04) and negatively correlated with balloons popped (r = −0.37; *p* = 0.04). For female participants, horizontal VOR negatively correlated with total money collected (r = −0.61; *p* = 0.02).

## 4. Discussion

This preliminary study is the first to investigate the relationship between action boundary perception, vestibular symptom provocation and impulsive behaviors following sport-related concussion in adolescents. In this cross-sectional study of 47 adolescents with sport-related concussion in the previous 21 days, weak-to-moderate strength associations were identified between inverse efficiency and accuracy during an assessment of action boundary perception and both vestibular symptom provocation and risk-taking behaviors. Biological sex appears to be a potential modifier of these relationships, as the number of statistically significant associations and the strength of those associations increased in recently concussed males when stratifying by sex but were not present in recently concussed females. These findings deepen our understanding of the multifactorial relationship between action boundary perception, intrinsic risk factors and post-concussion clinical presentation in adolescents and provide preliminary support for future investigations of biological sex as an important covariate to be considered.

Nearly all statistically significant associations with PACT outcomes were observed at the 1.2 ball–hole ratio, but not the 0.8 ball–hole ratio. This performance trend is notable as it indicates that both vestibular provocation and risk-taking behaviors were related to worse performance for an action that was afforded (i.e., possible to complete), but not an action that was not afforded. This result has important implications for subsequent injury risk following a concussion, as an inability to perceive an action that is not possible given the constraints of the environment could place an athlete in a position for a higher risk of injury. Prior research suggests that overestimating one’s action capabilities (i.e., perceiving that one can complete a certain action when it is not possible) is associated with a higher tendency toward accidents [20]. This suggests that clinicians should consider developing and using perceptual motor rehabilitation methods designed to reintegrate the concussed athlete into their environment prior to clearance for return to play [4,21,22]. Higher vestibular symptoms were associated with poorer efficiency scores on PACT at the 1.2 ratio, indicating that disruption of the vestibular system contributes to altered action boundary perception when the action is not afforded. Vestibular rehabilitation prior to returning to play could theoretically help recently concussed athletes re-attune their action boundary perception and limit subsequent injury risk [4].

Risk-taking behaviors have often been associated with a risk of concussion [23,24,25], but risk-taking has not been explored in relation to action boundary perception in a concussed population. In the present study, more variability in response time to pump the balloon (i.e., variability in how long it took to decide upon the risk as measured by the standard deviation of the response time) was associated with poorer accuracy at the 1.2 ratio on PACT. Higher response time variability was also associated with poorer efficiency at the 1.2 ratio on the PACT. These findings may suggest that an inconsistent risk-taking strategy is associated with poorer action boundary perception, but more research will be necessary to confirm this finding. Post-concussion vestibular symptoms were also associated with certain aspects of the BART, a finding which was magnified for male participants compared to female participants. Higher horizontal vestibular ocular reflex and visual motion sensitivity symptoms were associated with a more conservative risk-taking strategy in male participants, as evidenced by more balloons collected and fewer balloons popped. While not statistically significant, it may be notable that the reverse trend was observed in female participants. This is consistent with prior research which shows males tend to take more risks, especially in the context of a fiscal risk [26]. How biological sex alters risk-taking behaviors when experiencing vestibular symptoms is a novel area of research which requires future exploration.

### Limitations

There are limitations to this study including that it involved a secondary analysis of a cross-sectional study with a limited sample size. As such, the results presented, especially in the smaller female cohort, may be underpowered. We could not assess causal relationships between sport-related concussion, action boundary perception, vestibular symptoms and risk taking due to a single study timepoint. Participants were enrolled within 3 weeks of injury; a better understanding of longitudinal changes from the acute (24–72 h) to complete recovery timepoints would have value for the clinician aiming to restore action boundary perception prior to clearance for return to play. While the BART is considered a gold-standard assessment of risk-taking behavior, it may not completely account for physical risk-taking behaviors as it uses a simulated computerized task. The VOMS is not a complete vestibular evaluation, which often include methods such as caloric testing, a video head impulse test and vestibular evoked myogenic potentials. Future work should consider the relationship of PACT to these formal assessments.

## 5. Conclusions

Action boundary perception involves a multifactorial relationship between the individual and the environment, which appears to be dysregulated following a sport-related concussion. The results of this study suggest that a moderate strength association between risk-taking behaviors and vestibular symptoms/impairment with worse action boundary perception may exist. This relationship is more pronounced in male adolescents than females, in our sample, which may have been underpowered to look at differences by biological sex. These preliminary results have implications for return-to-play decision-making and future injury risk following sport-related concussion, as risk-taking behaviors are potentially modifiable and vestibular symptoms/impairment can be treated with targeted rehabilitation interventions.

## Figures and Tables

**Table 1 brainsci-15-00229-t001:** Descriptive statistics for the analyzed cohort (mean ± standard deviation).

	Overall (n = 47)	Male (n = 33)	Female (n = 14)	*p*
Age (Years)	15.2 ± 1.8	15.44 ± 1.66	14.50 ± 1.88	0.12
Biological Sex (Male)	33 (70.2%)	---	---	---
Days Since Concussion	9.5 ± 6.0	9.91 ± 6.41	8.36 ± 5.11	0.43
Number of Concussions	1.5 ± 0.8	1.64 ± 0.86	1.15 ± 0.56	0.03 *
ADD/ADHD History	3 (6.7%)	3 (9.7%)	0 (0.0%)	0.54
Motion Sickness History	12 (26.7%)	8 (25.8%)	4 (28.6%)	1.00
Migraine History	15 (33.3%)	11 (35.5%)	4 (28.6%)	0.74
Anxiety History	1 (2.2%)	0 (0.0%)	1 (7.1%)	0.31
Horizontal Vestibular Ocular Reflex	2.7 ± 3.9	2.9 ± 4.6	2.1 ± 2.0	0.55
Vertical Vestibular Ocular Reflex	2.7 ± 4.3	3.0 ± 4.9	1.9 ± 2.2	0.44
Visual Motion Sensitivity	3.9 ± 4.6	4.2 ± 5.2	3.3 ±2.8	0.55
PACT Accuracy at 0.8 Ratio	78.5 ± 13.1	78.28 ± 12.92	79.17 ± 13.84	0.84
PACT Accuracy at 1.2 Ratio	83.6 ± 13.2	82.64 ± 12.64	86.01 ± 14.65	0.43
PACT Response Time at 0.8 Ratio	0.98 ± 0.11	0.99 ± 0.11	0.98 ± 0.12	0.90
PACT Response Time at 1.2 Ratio	0.91 ± 0.12	0.91 ± 0.11	0.90 ± 0.12	0.81
PACT Inverse Efficiency at 0.8 Ratio	1.30 ± 0.32	1.30 ± 0.30	1.29 ± 0.37	0.95
PACT Inverse Efficiency at 1.2 Ratio	1.13 ± 0.34	1.14 ± 0.32	1.10 ± 0.39	0.70
BART Balloons Collected	17.98 ± 3.35	18.10 ± 3.64	17.71 ± 2.61	0.73
BART Balloons Popped	12.02 ± 3.35	11.91 ± 3.64	12.29 ± 2.61	0.73
BART Total Pumps	125.57 ± 26.65	123.85 ± 28.84	129.64 ± 20.97	0.50
BART Mean Response Time (ms)	866.86 ± 348.13	880.25 ± 373.69	835.29 ± 289.1	0.69
BART Standard Deviation Response Time (ms)	677.43 ± 353.13	704.25 ± 379.00	614.22 ± 285.54	0.43
BART Mean Pumps	4.19 ± 0.89	4.13 ± 0.96	4.32 ± 0.70	0.50
BART Standard Deviation of Pumps	2.12 ±0.54	2.10 ± 0.56	2.16 ± 0.51	0.74
BART Total Virtual Money Collected	96.43 ± 11.65	95.12 ± 12.14	99.5 ± 10.1	0.24

* statistically significant at *p* < 0.05; Abbreviations: ADD/ADHD = attention deficit disorder/attention deficit hyperactivity disorder, PACT = Perception–Action Coupling Task, BART = Balloon Analog Risk Task, ms = milliseconds.

**Table 2 brainsci-15-00229-t002:** Pearson correlation matrix between Perception–Action Coupling Task (PACT) outcomes with vestibular provocation symptoms on the Vestibular Ocular Motor Screen (VOMS) and Balloon Analog Risk Task (BART) outcomes among concussed adolescents (n = 47).

	ACC0.8	ACC1.2	RT0.8	RT1.2	IE0.8	IE1.2
HVOR	−0.11	−0.27	−0.03	0.21	0.11	0.33 *
VVOR	−0.15	−0.30	−0.03	0.17	0.13	0.34 *
VMS	−0.10	−0.30	−0.07	0.21	0.06	0.37 *
Balloons Collected	−0.11	−0.23	−0.15	0.03	0.02	0.20
Balloons Popped	0.11	0.23	0.15	−0.03	−0.02	−0.20
Total Pumps	0.07	0.20	0.23	0.07	0.04	−0.13
Mean RT Pumps (ms)	0.03	−0.10	−0.08	−0.06	−0.05	0.09
SD RT Pumps (ms)	−0.15	−0.47 **	0.06	0.09	0.16	0.42 **
Mean Pumps	0.07	0.20	0.23	0.07	0.04	−0.13
SD Pumps	−0.01	−0.09	0.34 *	0.21	0.15	0.13
Total Money Collected	−0.02	0.02	0.14	0.20	0.08	0.08

* statistically significant at *p* < 0.05; ** statistically significant at *p* < 0.01; Key: HVOR = horizontal vestibular ocular reflex, VVOR = vertical vestibular ocular reflex, RT = reaction time, SD = standard deviation, ms = milliseconds.

**Table 3 brainsci-15-00229-t003:** Pearson correlation coefficients between vestibular provocation symptoms on the Vestibular Ocular Motor Screen (VOMS) and Balloon Analog Risk Task (BART) outcomes among concussed adolescents (n = 47).

	HVOR	VVOR	VMS
Balloons Collected	0.30 *	0.25	0.25
Balloons Popped	−0.30 *	−0.25	−0.25
Total Pumps	−0.28	−0.25	−0.23
Mean RT Pumps (ms)	0.10	0.10	0.14
SD RT Pumps (ms)	0.18	0.20	0.26
Mean Pumps	−0.28	−0.25	−0.23
SD Pumps	−0.02	0.00	0.01
Total Money Collected	−0.06	−0.10	−0.07

* statistically significant at *p* < 0.05; Key: HVOR = horizontal vestibular ocular reflex, VVOR = vertical vestibular ocular reflex, RT = reaction time, SD = standard deviation, ms = milliseconds.

**Table 4 brainsci-15-00229-t004:** Pearson correlation coefficients between Perception–Action Coupling Task (PACT) outcomes with vestibular provocation symptoms on the Vestibular Ocular Motor Screen (VOMS) and Balloon Analog Risk Task (BART) outcomes.

		ACC0.8	ACC1.2	RT0.8	RT1.2	IE0.8	IE1.2
Male (n = 33)	HVOR	−0.15	−0.42 *	−0.07	0.23	0.14	0.47 **
	VVOR	−0.20	−0.42 *	−0.10	0.16	0.16	0.46 **
	VMS	−0.13	−0.44 *	−0.12	0.21	0.08	0.52 **
	Balloons Collected	−0.05	−0.36 *	−0.06	0.11	0.01	0.34
	Balloons Popped	0.05	0.36 *	0.06	−0.11	−0.01	−0.34
	Total Pumps	0.04	0.34	0.17	0.03	0.05	0.26
	Mean RT Pumps (ms)	0.12	−0.09	−0.05	−0.07	−0.15	−0.26
	SD RT Pumps (ms)	−0.06	−0.47 **	0.11	0.10	0.09	0.47 **
	Mean Pumps	0.04	0.34	0.17	0.03	0.05	−0.26
	SD Pumps	−0.17	−0.07	0.26	0.13	0.27	0.09
	Total Money Collected	0.07	0.05	0.20	0.27	0.04	0.07
Female (n = 14)	HVOR	0.09	0.36	0.14	0.15	−0.01	−0.23
	VVOR	0.07	0.21	0.25	0.24	0.06	−0.11
	VMS	0.07	0.21	0.12	0.25	−0.01	−0.14
	Balloons Collected	−0.29	0.15	−0.41	−0.20	0.05	−0.20
	Balloons Popped	0.29	−0.15	0.41	0.20	−0.05	0.20
	Total Pumps	0.17	−0.24	0.42	0.22	0.03	0.27
	Mean RT Pumps (ms)	−0.23	−0.11	−0.15	−0.04	0.20	0.03
	SD RT Pumps (ms)	−0.40	−0.46	−0.18	0.05	0.38	0.29
	Mean Pumps	0.17	−0.24	0.42	0.22	0.03	0.27
	SD Pumps	0.38	−0.14	0.52	0.43	−0.11	0.24
	Total Money Collected	−0.29	−0.15	0.00	0.06	0.18	0.14

* statistically significant at *p* < 0.05; ** statistically significant at *p* < 0.01; Key: HVOR = horizontal vestibular ocular reflex, VVOR = vertical vestibular ocular reflex, RT = reaction time, SD = standard deviation, ms = milliseconds.

**Table 5 brainsci-15-00229-t005:** Pearson correlation coefficients between vestibular provocation symptoms on the Vestibular Ocular Motor Screen (VOMS) and Balloon Analog Risk Task (BART) outcomes.

		HVOR	VVOR	VMS
Male (n = 33)	Balloons Collected	0.37 *	0.33	0.37 *
	Balloons Popped	−0.37 *	−0.33	−0.37 *
	Total Pumps	−0.30	−0.29	−0.31
	Mean RT Pumps (ms)	0.11	0.15	0.22
	SD RT Pumps (ms)	0.23	0.26	0.33
	Mean Pumps	−0.30	−0.29	−0.31
	SD Pumps	−0.05	−0.06	−0.09
	Total Money Collected	0.04	−0.02	0.02
Female (n = 14)	Balloons Collected	−0.18	−0.33	−0.47
	Balloons Popped	0.18	0.44	0.47
	Total Pumps	−0.10	0.10	0.29
	Mean RT Pumps (ms)	0.06	−0.19	−0.28
	SD RT Pumps (ms)	−0.19	−0.29	−0.19
	Mean Pumps	−0.10	0.11	0.29
	SD Pumps	0.13	0.32	0.46
	Total Money Collected	−0.61 *	−0.49	−0.44

* statistically significant at *p* < 0.05; Key: HVOR = horizontal vestibular ocular reflex, VVOR = vertical vestibular ocular reflex, RT = reaction time, SD = standard deviation, ms = milliseconds.

## Data Availability

The original contributions presented in the study are included in the article, further inquiries can be directed to the corresponding author.

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
