# Peer review of "Association of Risk-Taking Behaviors, Vestibular Provocation and Action Boundary Perception Following Sport-Related Concussion in Adolescents"

_brainsci, 2025, doi:10.3390/brainsci15030229_

Round 1
Reviewer 1 Report
Comments and Suggestions for Authors
First, I would like to thank you for providing me the opportunity to read this article. The current study examines risk-taking behaviours, vestibular symptoms, and action boundary perception following sports-related concussions. This topic has received little attention in research, making it a valuable area of investigation. However, some issues need to be addressed before this article can be considered for publication. Please refer to my specific recommendations outlined below.
Keywords
Using a more specific keyword instead of ’vestibular’ would be more beneficial.
Introduction
Considering vestibular dysfunction and concussion, it should be mentioned that vestibular dysfunction can be observed after concussion; however, pre-existing vestibular dysfunction can also increase the risk of concussion due to a greater likelihood of falls.
Line 46. It would be helpful to specify what is meant by ’vestibular symptoms.’
Introducing the term ’labyrinthine concussion’ could be beneficial for understanding concussion-related symptoms, specifically referring to sensorineural hearing loss, tinnitus, vertigo, and dizziness that occur after a head injury without any fractures of the labyrinth.
[Choi MS, Shin SO, Yeon JY, Choi YS, Kim J, Park SK. Clinical characteristics of labyrinthine concussion. Korean J Audiol. 2013 Apr;17(1):13-7. doi: 10.7874/kja.2013.17.1.13]
Line 61. Please correct it to either ’Research has shown’ or ’Researchers have shown.’
Material and Methods
Lines 75-76. It would be helpful to present the international consensus criteria for concussion in detail.
Line 81. Additionally, please provide the approval number and date of ethical approval. It is also necessary to include a statement regarding whether patients provided written informed consent.
Is there a specific type or manufacturer for the equipment used in PACT? If so, please provide the details.
To gain a better understanding of vestibular function and dysfunction, it is important to utilise objective vestibular testing methods such as caloric testing, the video head impulse test, and vestibular evoked myogenic potentials (VEMPs). These tests can be used to create a vestibulogram. The absence of these evaluations should be noted as a limitation and a consideration for future research.
Since the BART is a computerised test, all relevant details, including software and other specifics, should be provided.
In the context of statistical analysis, it is important to specify the name of the normality test used to assess the normal distribution. Additionally, it should be clarified that mean and SD values were employed for continuous variables. Furthermore, it should be noted that a significance level of p<0.05 was used, along with p<0.01.
Results
Table 1. It would be more effective to place the table caption above the table rather than inside it. Additionally, all abbreviations should be defined in the table caption. To enhance the scientific value of the data, it is important to not only collect parameters for males and females but also to perform statistical analyses to determine whether there are significant differences in the values between the two groups. Furthermore, an explanation of ’total money collected’ should be included, detailing its significance in relation to this investigation, as well as the currency used. A presentation of the results from this table should be included in the text, and the parameters should be collected with a clear explanation.
Lines 138-139. To begin with, it is important to clarify that the r-value indicates a significant positive correlation. Additionally, highlighting the strengths of these correlations based on the r-values would improve the presentation of the results. For example, r-values ranging from 0.33 to 0.37 represent moderate correlations. Additionally, to establish significant correlations, the p-values should be presented, for instance, in the table. Additionally, it is important to note any significant negative correlations that may exist, even including negative correlations.
Table 2. It would be more effective to place the table caption above the table rather than inside it. Additionally, all abbreviations should be defined in the table caption. It should be clarified that correlations were analysed using the Pearson correlation coefficient. Along with the r-values, the specific p-values must also be provided.
Lines 146-147. It should be clarified that based on the r-values these are significant negative correlations.
Table 3. It would be more effective to place the table caption above the table rather than inside it. Additionally, all abbreviations should be defined in the table caption. Along with the r-values, the specific p-values must also be provided.
Line 151. Please state that there is a negative correlation. Additionally, considering this subsection, the authors should provide practical explanations to enhance clarity, rather than solely collecting data in the text.
Table 4. It would be more effective to place the table caption above the table rather than inside it. Additionally, all abbreviations should be defined in the table caption. Along with the r-values, the specific p-values must also be provided.
Line 161. Positive and negative correlations should be specified.
Table 5. It would be more effective to place the table caption above the table rather than inside it. Additionally, all abbreviations should be defined in the table caption. Along with the r-values, the specific p-values must also be provided.
Discussion
Lines 202-203. It would be interesting if the authors could provide possible explanations for the results related to sex differences.
In light of study limitations, there is a need for more comprehensive vestibular testing, as previously stated.
I look forward to receiving the revised version of the manuscript.
Author Response
Reviewer #1
First, I would like to thank you for providing me the opportunity to read this article. The current study examines risk-taking behaviours, vestibular symptoms, and action boundary perception following sports-related concussions. This topic has received little attention in research, making it a valuable area of investigation. However, some issues need to be addressed before this article can be considered for publication. Please refer to my specific recommendations outlined below.
Keywords
REVIEWER COMMENT: Using a more specific keyword instead of ’vestibular’ would be more beneficial.
Author Response: This has been changed to vestibular symptoms/impairments.
Introduction
REVIEWER COMMENT: Considering vestibular dysfunction and concussion, it should be mentioned that vestibular dysfunction can be observed after concussion; however, pre-existing vestibular dysfunction can also increase the risk of concussion due to a greater likelihood of falls.
Author Response: The following has been added to the introduction: “As such, it is important to note that pre-existing vestibular dysfunction could be related to increased injury risk from a greater likelihood of falls.”
REVIEWER COMMENT: Line 46. It would be helpful to specify what is meant by ’vestibular symptoms.’
Author Response: Examples of vestibular symptoms are provided in the following paragraph: “Vestibular symptoms and impairment, such as dizziness, imbalance, vertigo, and gait dysfunction, which are among the most commonly affected domains following concussion…”
REVIEWER COMMENT: Introducing the term ’labyrinthine concussion’ could be beneficial for understanding concussion-related symptoms, specifically referring to sensorineural hearing loss, tinnitus, vertigo, and dizziness that occur after a head injury without any fractures of the labyrinth.
[Choi MS, Shin SO, Yeon JY, Choi YS, Kim J, Park SK. Clinical characteristics of labyrinthine concussion. Korean J Audiol. 2013 Apr;17(1):13-7. doi: 10.7874/kja.2013.17.1.13]
Author Response: the following has been added to the introduction: “The term “labyrinthine concussion” has been used to describe disruption to the vestibular system after head trauma.9”
REVIEWER COMMENT: Line 61. Please correct it to either ’Research has shown’ or ’Researchers have shown.’
Author Response: This has been corrected.
Material and Methods
REVIEWER COMMENT: Lines 75-76. It would be helpful to present the international consensus criteria for concussion in detail.
Author Response: The following has been added: “Sport-related concussion was diagnosed per international consensus criteria, which include reporting a mechanical force transmitted to the brain from either direct or indirect impacts resulting in signs and/or symptoms of brain injury.14”
REVIEWER COMMENT: Line 81. Additionally, please provide the approval number and date of ethical approval. It is also necessary to include a statement regarding whether patients provided written informed consent.
Author Response: This has been updated: “The University of Pittsburgh Institutional Review Board approved the study for human subjects research (ID number: PRO18060287; approved 9/6/2018). Participants provided written informed consent prior to particing in study procedures.”
REVIEWER COMMENT: Is there a specific type or manufacturer for the equipment used in PACT? If so, please provide the details.
Author Response: There is not a specific type or manufacturer for equipment.
REVIEWER COMMENT: To gain a better understanding of vestibular function and dysfunction, it is important to utilise objective vestibular testing methods such as caloric testing, the video head impulse test, and vestibular evoked myogenic potentials (VEMPs). These tests can be used to create a vestibulogram. The absence of these evaluations should be noted as a limitation and a consideration for future research.
Author Response: The following has been added to the limitations: “The VOMS is not a complete vestibular evaluation, which often include methods such as caloric testing, a video head impulse test and vestibular evoked myogenic potentials. Future work should consider the relationship of PACT to these formal assessments.”
REVIEWER COMMENT: Since the BART is a computerised test, all relevant details, including software and other specifics, should be provided.
Author Response: The following has been added to the methods: “(Psychology Software Tools, Inc.; Sharpsburg, Pennsylvania).”
REVIEWER COMMENT: In the context of statistical analysis, it is important to specify the name of the normality test used to assess the normal distribution. Additionally, it should be clarified that mean and SD values were employed for continuous variables. Furthermore, it should be noted that a significance level of p<0.05 was used, along with p<0.01.
Author Response: This has been edited: “Data were assessed for normality using the Shapiro-Wilk test. Descriptive statistics were calculated for all outcome measures, including means and standard deviations for continuous variables. Pearson correlation matrices were developed to evaluate the association between BART, VOMS and PACT outcomes within the study cohort of concussed adolescents. Because male adolescents take risks more frequently than female adolescents, correlation matrices were also stratified by biological sex assigned at birth. Alpha level was set a priori to p<0.05 but those with p<0.01 are also reported.”
Results
REVIEWER COMMENT: Table 1. It would be more effective to place the table caption above the table rather than inside it. Additionally, all abbreviations should be defined in the table caption. To enhance the scientific value of the data, it is important to not only collect parameters for males and females but also to perform statistical analyses to determine whether there are significant differences in the values between the two groups. Furthermore, an explanation of ’total money collected’ should be included, detailing its significance in relation to this investigation, as well as the currency used. A presentation of the results from this table should be included in the text, and the parameters should be collected with a clear explanation.
Author Response: These parameters have been added. Total money collected is an outcome of BART, which has been more clearly labelled.
REVIEWER COMMENT: Lines 138-139. To begin with, it is important to clarify that the r-value indicates a significant positive correlation. Additionally, highlighting the strengths of these correlations based on the r-values would improve the presentation of the results. For example, r-values ranging from 0.33 to 0.37 represent moderate correlations. Additionally, to establish significant correlations, the p-values should be presented, for instance, in the table. Additionally, it is important to note any significant negative correlations that may exist, even including negative correlations.
Author Response: Positive and negative status has been explicitly stated in the results section. We also added the following to the Statistical Analysis section to aid in interpretation: “Correlation strength was interpreted as 0.10-0.39=weak correlation, 0.40-0.69 as moderate correlation, 0.70-0.89 as strong correlation, and 0.90-1.00 as very strong correlation.18”
P-values for significant correlations have been reported in the Results section, but not in the tables to reduce visual clutter and redundancy.
REVIEWER COMMENT: Table 2. It would be more effective to place the table caption above the table rather than inside it. Additionally, all abbreviations should be defined in the table caption. It should be clarified that correlations were analysed using the Pearson correlation coefficient. Along with the r-values, the specific p-values must also be provided.
Author Response: The table caption has been moved above and abbreviations added. Pearson correlation was already named in the title of the table.
REVIEWER COMMENT: Lines 146-147. It should be clarified that based on the r-values these are significant negative correlations.
Author Response: This has been clarified.
REVIEWER COMMENT: Table 3. It would be more effective to place the table caption above the table rather than inside it. Additionally, all abbreviations should be defined in the table caption. Along with the r-values, the specific p-values must also be provided.
Author Response: Done.
REVIEWER COMMENT: Line 151. Please state that there is a negative correlation. Additionally, considering this subsection, the authors should provide practical explanations to enhance clarity, rather than solely collecting data in the text.
Author Response: Done.
REVIEWER COMMENT: Table 4. It would be more effective to place the table caption above the table rather than inside it. Additionally, all abbreviations should be defined in the table caption. Along with the r-values, the specific p-values must also be provided.
Author Response: Done.
REVIEWER COMMENT: Line 161. Positive and negative correlations should be specified.
Author Response: Done.
REVIEWER COMMENT: Table 5. It would be more effective to place the table caption above the table rather than inside it. Additionally, all abbreviations should be defined in the table caption. Along with the r-values, the specific p-values must also be provided.
Author Response: Done.
Discussion
REVIEWER COMMENT: Lines 202-203. It would be interesting if the authors could provide possible explanations for the results related to sex differences.
Author Response: The following has been added to the discussion: “This is consistent with prior research which shows males tend to take more risks, especially in the context of a fiscal risk.21” Because this is a novel and multifactorial result, we do not want to offer further exploration to this preliminary study based upon potentially unfounded conclusions, as this was a secondary analysis of the study presented.
REVIEWER COMMENT: In light of study limitations, there is a need for more comprehensive vestibular testing, as previously stated.
Author Response: We added this as a limitation in response to your above comment.
Reviewer 2 Report
Comments and Suggestions for Authors
Abstract
Comments for the authors
Introduction
· Line 42-43: It is not clear what does “return from concussion and musculoskeletal injury” mean. Does this mean functional recovery? Please explain.
· Line 48-51: Please provide additional information on how altered perception-action coupling leads to increased injury risk.
· Line 56-57: The sentence says “these impulsivity measures”. Were there other measure too that reported findings similar to Barratt Impulsivity scale? Also, what was the strength of association (weak, moderate, strong?
· Line 56-57: Please double check the name “Barratt Impulsivity scale”. I think the correct terms is Barratt Impulsiveness scale.
· The theoretical basis for the need to link vestibular symptoms with perception-action coupling is unclear. Please provide additional information on this so the reader can understand why these two constructs should be explored for a relationship. How will this information be helpful to the clinicians?
· The way hypothesis # 2 (vestibular symptoms) is currently written, it suggests a cause-effect relationship rather than association. Please revise.
Methods
· Please include the ethical board approval number.
· Was the sample size calculated a-priori? Please provide details.
· Since it is essential that the measures demonstrate population specific validity, please add a description of the psychometric properties for all the measures that were used in this study.
· What is inverse efficiency? Please explain.
· Please provide details on the scoring interpretation for BART i.e. what does a high/low score mean?
· Was the data assessed for normality? Please explain.
· Which guidelines were used for interpretation of the correlation coefficient? Please provide a citation.
· Please provide a citation that supports that statement that “male adolescents take risks more frequently than female adolescents”.
Results
· Please make the tables stand alone.
· Why was standard deviation of BART correlated instead of mean? Please elaborate.
· In table 3 and 5, there was no value that had a p < 0.001. Please correct this.
· Why were correlation values were calculated separately for males and females? A sample of 14 appears too small and may not provide meaningful results. Please explain.
· Please comment on the strength of association observed between the variables and what is the inference that can be drawn from these?
Discussion
Overall, the discussion is weak, choppy and does not flow well. Most of the section seems like a repetition of the results.
Line 171-173: The statistical analysis only indicates an association. How was the determination made that biological sex is a modifier? Please explain.
Line 180-183: This segment is unclear. How is this relevant for a clinician?
Author Response
Reviewer #2
REVIEWER COMMENT: Line 42-43: It is not clear what does “return from concussion and musculoskeletal injury” mean. Does this mean functional recovery? Please explain.
Author Response: This has been clarified: “No clear causal mechanism has emerged linking concussion and subsequent musculoskeletal injury within the first few months of returning to activity, but poor sensorimotor control and/or premature return to activity from concussion have been postulated”
REVIEWER COMMENT: Line 48-51: Please provide additional information on how altered perception-action coupling leads to increased injury risk.
Author Response: The following has been added to this section: “Trauma to the head could alter a person’s perceptions of their environment and/or their attunement to their own abilities, both of which could factor into increasing subsequent injury risk.4”
REVIEWER COMMENT: Line 56-57: The sentence says “these impulsivity measures”. Were there other measure too that reported findings similar to Barratt Impulsivity scale? Also, what was the strength of association (weak, moderate, strong?
Author Response: The following has been updated: “…cognitive instability and attention impulsivity were associated with perception-action coupling performance in multivariable modelling amongst the concussed adolescents”. We did not report strength of association because we are referring to regression models and there is no established criteria for reporting strength of coefficients in these models.
REVIEWER COMMENT: Line 56-57: Please double check the name “Barratt Impulsivity scale”. I think the correct terms is Barratt Impulsiveness scale.
Author Response: Thank you. This has been changed to Impulsiveness.
REVIEWER COMMENT: The theoretical basis for the need to link vestibular symptoms with perception-action coupling is unclear. Please provide additional information on this so the reader can understand why these two constructs should be explored for a relationship. How will this information be helpful to the clinicians?
Author Response: The following has been added:
“Trauma to the head could alter a person’s perceptions of their environment and/or their attunement to their own abilities, both of which could factor into increasing subsequent injury risk.4”
“This suggests that clinicians should consider developing and using perceptual motor rehabilitation methods designed to reintegrate the concussed athlete to their environment prior to clearance for return to play.4”
REVIEWER COMMENT: The way hypothesis # 2 (vestibular symptoms) is currently written, it suggests a cause-effect relationship rather than association. Please revise.
Author Response: This has been edited: “We also hypothesize that higher vestibular symptoms will be associated with worse perception action coupling performance.”
Methods
REVIEWER COMMENT: Please include the ethical board approval number.
Author Response: This has been added.
REVIEWER COMMENT: Was the sample size calculated a-priori? Please provide details.
Author Response: The following has been added to clarify: “This is a secondary analysis for which the original study was not statistically powered a priori.” This is also mentioned in the limitations.
REVIEWER COMMENT: Since it is essential that the measures demonstrate population specific validity, please add a description of the psychometric properties for all the measures that were used in this study.
Author Response: We have added a sentence to each outcome to each reliability and population validity:
“The PACT has good intersession reliability (intraclass correlation coefficients=0.78-0.94) and within-subject variability,15 and population validity in patients with concussion.8,16”
“The VOMS has good test-retest reliability (0.60-0.81)18 and internal consistency (Cronbach alpha = 0.97)19with population validity in the concussed patient.20-22”
“The BART has good test-retest reliability (intraclass correlation coefficient = 0.77)24 and population validity in participants with concussion.25”
REVIEWER COMMENT: What is inverse efficiency? Please explain.
Author Response: Inverse efficiency is defined in the PACT paragraph: “Inverse efficiency is the combination of speed and accuracy into a single measure where higher scores indicate a less efficient performance (i.e., slower responses and more errors) and lower scores indicate more efficient performances (i.e., fasters responses and accurate responses).”
REVIEWER COMMENT: Please provide details on the scoring interpretation for BART i.e. what does a high/low score mean?
Author Response: We have added the following to the methods: “More balloons collected/fewer pumps reflects a more conservative strategy, whereas more balloons popped/more pumps reflects a more aggressive strategy.”
REVIEWER COMMENT: Was the data assessed for normality? Please explain.
Author Response: Yes, the following has been added: “Data were assessed for normality using the Shapiro-Wilk test.”
REVIEWER COMMENT: Which guidelines were used for interpretation of the correlation coefficient? Please provide a citation.
Author Response: This has been added: “Correlation strength was interpreted as 0.10-0.39=weak correlation, 0.40-0.69 as moderate correlation, 0.70-0.89 as strong correlation, and 0.90-1.00 as very strong correlation.18”
REVIEWER COMMENT: Please provide a citation that supports that statement that “male adolescents take risks more frequently than female adolescents”.
Author Response: Done.
Results
REVIEWER COMMENT: Please make the tables stand alone.
Author Response: Done.
REVIEWER COMMENT: Why was standard deviation of BART correlated instead of mean? Please elaborate.
Author Response: This has been clarified in the discussion: “In the present study, more variability in response time to pump the balloon (i.e., variability in how long it took to decide upon the risk as measured by the standard deviation of response time) was associated with poorer accuracy at the 1.2 ratio on PACT. Higher response time variability was also associated with poorer efficiency at the 1.2 ratio on the PACT. These findings may suggest that an inconsistent risk-taking strategy is associated with poorer action boundary perception, but more research will be necessary to confirm this finding.” Because this is a preliminary finding that is potentially underpowered, we do not want to overinterpret it.
REVIEWER COMMENT: In table 3 and 5, there was no value that had a p < 0.001. Please correct this.
Author Response: Thank you- this option has been removed.
REVIEWER COMMENT: Why were correlation values were calculated separately for males and females? A sample of 14 appears too small and may not provide meaningful results. Please explain.
Author Response: They were separated because of notable differences in male and female adolescents in both risk taking and concussion response, which we point out in the introduction, methods, and discussion. The sample size is admittedly small but we disclose multiple times in the discussion/limitations that the sample sizes were small and potentially underpowered: “As such, the results presented, especially in the smaller female cohort, may be underpowered.”
REVIEWER COMMENT: Please comment on the strength of association observed between the variables and what is the inference that can be drawn from these?
Author Response: We have added strength interpretations to the Methods section, and explicitly mention the strength of the associations now in the Discussion and Conclusion:
“In this cross-sectional study of 47 adolescents with sport-related concussion in the previous 21 days, weak-to-moderate strength associations were identified…”
“The results of this study suggest that a moderate strength association between risk-taking behaviors and vestibular symptoms/impairment with worse action boundary perception may exist. This relationship is more pronounced in male adolescents than females, in our sample, which may have been underpowered to look at differences by biological sex.”
REVIEWER COMMENT: Line 171-173: The statistical analysis only indicates an association. How was the determination made that biological sex is a modifier? Please explain.
Author Response: The following provides that explanation, but the word significant has been replaced with the word potential to clarify our intent as a preliminary finding: “Biological sex appears to be a potential modifier of these relationships, as the number of statistically significant associations and the strength of those associations increased in recently concussed males when stratifying by sex but were not present in recently concussed females.”
REVIEWER COMMENT: Line 180-183: This segment is unclear. How is this relevant for a clinician?
Author Response: The following has been added to clarify: “This suggests that clinicians should consider developing and using perceptual motor rehabilitation methods designed to reintegrate the concussed athlete to their environment prior to clearance for return to play”
Round 2
Reviewer 1 Report
Comments and Suggestions for Authors
Thank you for sending me the revised version of the manuscript. The authors have made significant efforts to enhance its quality, addressing the most critical corrections effectively. Therefore, in my opinion, it is now ready to be considered for publication.